# Occurrence and Ecological Risks of Neonicotinoids in Wheat, Corn and Rice Field Soils in China

**DOI:** 10.3390/molecules30081803

**Published:** 2025-04-17

**Authors:** Junxue Wu, Pingzhong Yu, Ziyu Zou, Ercheng Zhao, Junjie Jing, Jinwei Zhang, Yan Tao, Lirui Ren, Min He, Li Chen, Ping Han

**Affiliations:** Beijing Key Laboratory of Environment Friendly Management on Fruit Diseases and Pests in North China, Key Laboratory of Environment Friendly Management on Fruit and Vegetable Pests in North China (Co-Construction by Ministry and Province), Ministry of Agriculture and Rural Affairs, Institute of Plant Protection, Beijing Academy of Agriculture and Forestry Sciences, Beijing 100097, China; wujunxue@baafs.net.cn (J.W.); yupingzhong@baafs.net.cn (P.Y.); ziyu2965@163.com (Z.Z.); jingjunjie@baafs.net.cn (J.J.); zhangjinwei@baafs.net.cn (J.Z.); taoyan@baafs.net.cn (Y.T.); renlirui@baafs.net.cn (L.R.); hemin@baafs.net.cn (M.H.); chenli@baafs.net.cn (L.C.)

**Keywords:** neonicotinoids, occurrence, soil, ecological risks

## Abstract

The global application of neonicotinoids (NEOs) has precipitated pervasive contamination of agricultural matrices, with China’s staple crop lands representing critical exposure hotspots. The occurrence and ecological risks of ten NEOs in the field soils of three major crops (i.e., rice, wheat and corn) in China were investigated in the present study. Employing an optimized UPLC-MS/MS method (LOQ = 0.01–1.7 ng/g, RSD < 12.21%), ten NEOs across 69 representative field soils (rice: 23, corn: 18, wheat: 28) were quantified. It was found that the detection frequency (DF) of the NEOs was 100% in the soil. The DFs of NEOs in the soil followed the rule: imidacloprid (IMI, 100%) > thiamethoxam (TMX, 88.4%) > clothianidin (CLO, 87.0%) > acetamiprid (ACE, 46.4%) > dinotefuran (DIN, 7.2%) > nitenpyram (NIT, 1.4%). Mean total detected NEOs concentrations exhibited crop-dependent type: wheat (1.77–214.55 ng/g) > corn (0.79–97.53 ng/g) > rice (0.75–72.97 ng/g). The IMI, CLO and TMX triad constituted over 90% of the total contribution of detected NEOs. In addition, CLO and TMX in the rice soils, IMI, CLO and TMX in the corn soils and IMI, CLO and TMX in the wheat soils had medium ecological risks. Therefore, it is particularly important for agricultural ecological protection to strengthen monitoring and take effective measures to protect agricultural ecology.

## 1. Introduction

Neonicotinoids (NEOs) are one of the most effective insecticides for controlling stinging insects, small lepidoptera and coleoptera pests [1]. Action mechanisms of the NEOs are nerve stimulation at low concentrations, blockage of nicotinic acetylcholine receptors, overactive neurons of insects to paralysis and even death at high concentrations [2]. Due to the broad spectrum and no cross resistance, NEOs account for over 25% of the global pesticide market and have been widely applied in more than 120 countries, with an annual expenditure of $1.9 billion all over the world [3]. Years of NEOs application might have led to environmental pollution and ecological health risks. Understanding the pollution level of NEOs is a first and important step in managing and reducing their pollution in the environment.

Referring to research on NEOs pollution, it is found that NEOs have been frequently determined in environmental matrices such as water, soil and sediments [4,5,6,7]. For example, imidacloprid (IMI) has been detected in the surface stream water of the Cachapoal River basin (central Chile) [8], the Basque coast (N Spain) [9] and the surface water of Taihu Lake (E China) [10]. The IMI concentrations exceeded ecotoxicity limits for aquatic invertebrates in lake waters [11]. Bonmatin et al. [12] reported that the detection frequency (DF) of NEOs was higher in soil (68%) than in water samples (12%), and the total NEOs concentrations in the soil were 10 times higher than those in field sediments (0.014–0.348 ng/g). DFs of IMI reached 91% in French planting areas, and residues of IMI were obviously higher in plots sown for two consecutive years than those in one-year planting areas [13] and might tend to be stable after 3–4 years of repeated annual use in agricultural soils [14]. Based on reports of NEOs pollution in soil in Belize [12], Philippines [15], southwestern Ontario [14], England [16] and Ghana [17], NEOs residues were persistent and generally existed in the concentration level of ng~μg/g. Long-term exposure to NEOs would have adverse effects on non-target terrestrial arthropods and aquatic invertebrates [18,19]. As a result, the pollution level and risk assessment of NEOs have attracted much more attention regarding the environment.

China is a typical large agricultural country with a large area of cultivated land and developed planting industry. NEOs are widely produced and used in China, especially in agricultural areas. So far, NEOs have been reported to be detected in surface water such as in the Yangtze River and Pearl River, but the data about the occurrence and distribution of NEOs in field soils in China are limited. Chinese rice, corn and wheat are three major crops accounting for a high market demand all year round. IMI products are often used for spraying, and clothianidin (CLO) and thiamethoxam (TMX) products are often used as seed coating agents in the agricultural practice of the three crops. Despite the superior climate and natural geographical conditions, harvesting of Chinese rice, wheat and corn cannot be achieved without the use of insecticides. Hence, the exposure level and risks of NEOs in field soils are worthy of being further revealed.

In the present study, the occurrence and ecological risks of ten NEOs in the field soils of three major crops (i.e., rice, wheat and corn) in China were investigated. The main objectives of this study are (1) to investigate the pollution levels of NEOs in soils of three major crops in China, (2) to explore the potential differences of NEOs residues in different regional and planted crop soils and (3) to evaluate the potential environmental ecological risks of NEOs. This study will provide a theoretical basis for paying more attention to the pollution and risk of NEOs in soil ecosystems.

## 2. Results and Discussion

### 2.1. Occurrence of NEOs in Soil Samples

DFs and concentrations (maximum, minimum, median and mean) of ten NEOs in the rice, corn and wheat field soil samples are listed in Table 1. Results indicated that no less than one NEO could be detected in all the soil samples, thereby the DFs of the NEOs were 100%. Among all the soil samples, IMI had high detection frequencies (100%), followed by TMX (88.4%), CLO (87.0%), ACE (46.4%), DIN (7.2%) and NIT (1.4%). Five NEOs were detected in the rice soil, which were IMI (100%), TMX (87.0%), CLO (78.3%), ACE (56.5%) and DIN (13.0%), respectively. For the corn soil, both IMI and TMX were found with 100% detection frequencies, followed by CLO (83.3%) and ACE (38.9%). Similarly to rice and corn soil, IMI in the wheat field soil was also 100% detected. Besides IMI, CLO (96.4%), TMX (82.1%), ACE (42.9%), DIN (7.1%) and NIT (3.6%) were also found in the wheat soil. The other four NEOs (i.e., THIA, IMTH, CYC and PCD) were not detected in the soil samples. DIN was only detected in a small number of soil samples from rice and wheat fields, but not in corn fields. NIT was only detected in one soil sample from a wheat field. The results therefore show that the top four most widely used NEOs were IMI, CLO, TMX and ACE in the fields in China.

As described in Figure 1, the total detected concentrations of the NEOs reached 0.75–73.0, 0.79–97.5 and 1.77–214 ng/g in the rice, corn and wheat soils, respectively. As listed in Table 1, among the detected NEOs, IMI had the highest mean (or median) concentrations of 5.21 (0.41), 12.78 (8.94) and 109.61 (8.32) ng/g in the rice, corn and wheat soils, respectively. The possible reasons for this phenomenon were consumer preference and multi-purpose patterns of IMI products [20]. As shown in Figure 2, IMI accounted for 2.9–64.6%, 4.4–96.2% and 2.4–34.1% in the rice soils from HN, JS and LN provinces, 10.8–97.2%, 44.4–86.6% and 36.3–59.0% in the corn soils from LN, SD and HB provinces and 17.4–87.7%, 15.1–70.5% and 1.9–95.2% in the wheat soils from JS and SD provinces and BJ city, respectively. High DFs and concentration levels of IMI indicated that IMI had a wider application in Chinese crop fields. Concentrations of IMI, CLO, TMX, ACE and DIN were in the range of 0.18–70.20, 0.19–30.09, 0.48–21.63, <LOQ-1.66 and <LOQ-5.81 ng/g in the rice soil, respectively (Figure 1a). The contribution rates of IMI, CLO, TMX, ACE and DIN in the rice soil were 96.2%, 97.1%, 71.1%, 61.6% and 49.9%, respectively (Figure 2a). For the corn field soils, the maximum detected concentrations of IMI, CLO, TMX and ACE were up to 50.75, 29.88, 16.90 and 1.05 ng/g with contribution rates of 97.2%, 43.5%, 75.6% and 10.3%, respectively (Figure 1b and Figure 2b). In the wheat soils, the maximum detected concentrations of NEOs followed the order IMI (155.70 ng/g) > TMX (79.39 ng/g) > CLO (43.46 ng/g) > ACE (8.10 ng/g) > DIN (7.58 ng/g) > NIT (4.11 ng/g), with contribution percentages of 95.2%, 90.3%, 64.3%, 48.0%, 35.0% and 29.2%, respectively (Figure 1c and Figure 2c).

As shown in Figure 2, mean total concentrations of the detected NEOs were compared in soil samples from different provinces and crops. The mean total concentrations of the detected NEOs in the soils from different provinces followed: JS (27.4 ng/g) > HN (8.7 ng/g) > LN (3.3 ng/g) in the rice soil, HB (43.5 ng/g) > SD (32.2 ng/g) > LN (8.4 ng/g) in the corn soil and SD (67.1 ng/g) > BJ (44.2 ng/g) > JS (28.7 ng/g) in the wheat soil. Generally, elevated soil temperatures enhance microbial activity and chemical reaction rates, thereby accelerating pesticide degradation. Adequate soil moisture facilitates microbial proliferation and pesticide dissolution, promoting breakdown processes. However, waterlogged conditions may induce anaerobic environments that alter degradation pathways. While higher organic matter content typically increases pesticide adsorption and reduces bioavailability, it may conversely stimulate microbial activity under optimal conditions, potentially expediting decomposition. Climatically favorable regions in southern China (e.g., HN and JS) with warm or humid conditions would theoretically exhibit enhanced pesticide degradation. Nevertheless, this study reveals significantly lower NEOs residues in northern soils (e.g., LN) compared to southern counterparts. The observed higher residues in the south likely result from intensive pesticide use, frequent pest outbreaks and/or soil conditions that slow degradation despite favorable climate factors. The application frequency of pesticides has a close relationship with the occurrence pattern of insect pests during plant growing, resulting in different pesticide pollution situations in the field soils. LN province is not conducive to the benefits for the reproduction of pests compared with warm or humid conditions in southern China, resulting in less application of the NEOs, which might be one reason for the low concentration levels of the detected NEOs in the soils from LN province.

In the same province, the mean total concentrations of the detected NEOs in the wheat soil were higher than those in the corn soil in the SD and JS provinces and in the corn soil were higher than those in the rice soil in the LN provinces. As shown in Figure 3, the mean total concentrations of the detected NEOs in the collected soils from different crops followed: wheat (42.47 ng/g) > corn (22.08 ng/g) > rice (12.05 ng/g). The composition distribution of the detected NEOs in the different provinces and crop types of soils is shown in Figure 4. It indicated that three NEOs (i.e., IMI, CLO and TMX) accounted for 93.6–94.4% (mean, 94.1%), 97.2–100% (mean, 99.5%) and 94.1–99.6% (mean, 97.4%) in the rice, corn and wheat soil, respectively. The ACE with more than 40% detection rates only contributed 2.2%, 0.5% and 1.6% for the total concentrations of NEOs in the rice, corn and wheat soil. Considering the crop types, the mean contribution percentages of the detected IMI followed rice soil (43.2%) < corn soil (47.8%) < wheat soil (58.6%), the detected CLO followed rice soil (33.3%) > corn soil (22.6%) > wheat soil (13.8%) and the detected TMX followed rice soil (17.6%) < corn soil (19.1%) < wheat soil (24.9%), respectively. The results indicated that the total concentrations of NEOs was mainly composed of IMI, CLO, and TMX in the soils.

Comparisons of the detected NEOs in soil samples around the world are summarized in Table 2. It indicates that crop species and regions have significant impacts on the occurrence of NEOs in field soils. Furthermore, the detected NEOs had a similar composition (mainly consisting of IMI, TMX, CLO and ACE) with different concentration levels. The concentration levels of the detected NEOs were in the range of ng-μg g^−1^ in global field soils. The concentration of the detected NEOs in the crop soil were lower than that in the vegetable soil and higher than that in the citrus orchard soil in China [5,21,22,23]. The concentration levels of the detected NEOs in Chinese field soils were higher than those in field soils in England, Canada, Belize and the Philippines [12,14,15,16]. According to the reported literature, it was also found that the concentrations of the detected NEOs generally followed a rule: vegetable soil > crop soil > orchard soil > fallow soil. The above indicates that NEOs have varying degrees of pollution in global field soils.

### 2.2. Ecological Risk Assessment

Ecotoxicological risk of the detected NEOs in the soil environment was assessed using the RQ method. In the present study, earthworms, *Folsomia candida* and *Hypoaspis aculeifer* were chosen as the ecological risk assessment object in the soil environment to assess the ecological risk of the exposure of NEOs. The ecological risk of the detected NEOs had a very close relationship with their concentrations in the soil and PNEC values. As shown in Figure 5, the median values of the RQ of IMI, CLO, TMX, DIN and ACE were 0.02, 0.21, 0.16, 0 and 0 in the rice soils, 0.50, 0.76, 0.26, 0 and 0 in the corn soils and 0.47, 0.87, 0.40, 0 and 0 in the wheat soils, respectively. This indicated that CLO and TMX in rice soils, IMI, CLO and TMX in corn soils and IMI, CLO and TMX in wheat soils would have a medium ecological risk with 0.1 < RQ < 1. A low ecological risk with a RQ < 0.1 was found for other detected NEOs in the soils. The max RQ values of the detected IMI (8.75), CLO (22.87), TMX (20.89) and DIN (16.85) were higher than 1 in the soils, indicating that the high ecological risk of single detected NEOs cannot be ignored. Yu et al. [24] found that the current residual levels of NEOs in the soils of an agricultural zone within the Pearl River Delta in South China could pose sub-lethal or acute effects to non-target terrestrial organisms such as earthworms. The application of IMI in peanut fields treated with flowable concentrate for seed treatment formulations also predicted low to moderate earthworm toxicity and a medium risk from ecotoxicity exposure [25]. Wu et al. [26] reported that the ecological risks of ACE and TMX were higher than other NEOs in the soils of tomato and cucumber greenhouses. Consequently, the ecological risks posed by pesticide residues in agricultural soils exhibit significant variation across different cropping systems. This underscores the critical need for enhanced environmental monitoring, with particular emphasis on NEOs in soil ecosystems.

## 3. Methods

### 3.1. Chemical Reagents

The standards of ten NEOs (IMI; CLO; TMX; dinotefuran, DIN; acetamiprid, ACE; thiacloprid, THIA; imidaclothiz, IMTH; nitenpyram, NIT; cycloxaprid, CYC; paichongding, PCD) were purchased from J&K Scientific Ltd., Shanghai, China. The basic information including the physical and chemical properties (p*K*_a_, K_ow_) are listed in Appendix A. Chromatographic grades of acetonitrile (ACN) and methanol (MeOH) used in the experimental analysis were purchased from Honeywell trading Shanghai Co., Ltd., Shanghai, China. Sodium chloride (NaCl) and formic acid (FA) were obtained from Sinopharm Chemical Reagent Co. Ltd., Shanghai, China. Purification adsorbent bondesil primary secondary amine (PSA, 40–60 μm) was obtained from Agela Technologies, Tianjin, China. Ultrapure water was prepared with a Millipore Milli-Q system (Millipore, Milford, MA, USA).

A total of 69 topsoil samples (0–20 cm) from the planting areas of three typical crops (rice, corn and wheat) were collected in six provinces of China in 2021. Specifically, 23 soil samples from rice growing areas in Liaoning (LN), Hai nan (HN) and Jiangsu (JS) provinces, 18 soil samples from corn fields in LN, Shandong (SD) and Hebei (HB) provinces and 28 soil samples from wheat farmland in SD, JS and Beijing (BJ) were collected. The collected soil samples were mixed well after removing solid impurities (stones, plant debris, etc.), ground through a 2 mm sieve after freeze-drying and stored a freezer at −20 °C.

### 3.2. Sample Extraction and Analysis

The sample extraction method is referred to by Zou et al. [27]. Specifically, a 5.0 g soil sample was weighed into a 50 mL centrifuge tube, and 5 mL of ultrapure water, 5 mL of ACN and 2 g of NaCl were added, vortexed on a Multi-Tube Vortexer at 2500 rpm for 5 min and centrifuged at 4000× *g* rpm for 5 min. Then, 1 mL of supernatant was added into a 2 mL centrifuge tube containing 50 mg of PSA, vortexed on a Vortexer at 2500 rpm for 3 min and centrifuged at 10,000× *g* rpm for 2 min. The sample was analyzed after filtered with a 0.22 μm PTFE filter.

The concentrations of ten NEOs were determined via ultra-high performance liquid chromatography triple quadrupole tandem mass spectrometry (UPLC-MS/MS, Waters Corp, Milford, MA, USA) with wide detection, and high sensitivity was used for the determination of the NEOs. The analytes were separated on a Waters ACQUITY UPLC BEH C18 column (1.7 μm, 100 mm × 2.1 mm), and the mobile phase contained 0.1% FA (A) and ACN (B) at a flow rate of 0.25 mL/min. The gradient elution program was performed as follows: 0–3.0 min, 10–80% B; 3.0–4.0 min, 80% B; 4.0–6.0 min, 80–10% B; 6.0–8.0 min, 10% B. The column temperature was maintained at 30 °C. The injection volume was 5 μL. The instrument operates in positive electrospray ionization (ESI+) with the following parameters: desolvation gas temperature 350 °C, source temperature 150 °C, capillary voltage 3.5 kV, desolvation gas flow 550 L/h and cone gas flow 50 L/h. The multiple reaction monitoring (MRM) data acquisition parameters of ten NEOs are detailed in Table 3.

### 3.3. Quality Assurance and Quality Control

The analysis experiment was conducted by setting sample blank, operation blank and blank spiked recovery. A set of quality control samples was set after each 10 samples to ensure the feasibility of the method. The blank soil samples were collected from the experiment field of the Beijing Academy of Agriculture and Forestry Sciences, which has no history of target pesticides application. Matrix spike recoveries of ten NEOs at three levels in the soils are 61.8–124.2%, and the relative standard deviations are 0.48–12.21% (*n* = 3). The linearity of ten NEOs in a range of 0.1–100 ng/g was evaluated using matrix standard curves with a good linear relationship (R > 0.99). The recovery and linearity results are detailed in Appendix A. The limits of detection (LOD) and limits of quantitation (LOQ) of ten NEOs were calculated with 3 times and 10 times signal-to-noise ratio, respectively. The LODs and LOQs of the 10 NEOs were in the range of 0.003–0.5 and 0.01–1.7 ng/g. Concentrations of analytes below LOQs were labeled as non-detected (ND), and concentrations greater than or equal to the LODs but less than the LOQs were assigned the LOQ/2 value for statistical analysis.

### 3.4. Ecological Risk Assessment

The risk quotient (RQ) method can be used to analyze the potential ecological risks of the NEOs for soil non-target organisms [28,29]. The RQ values were calculated using Equation (1). Ecological risk levels can be defined as follows: high risk (RQ ≥ 1), medium risk (0.1 ≤ RQ ≤ 1) and low risk (RQ ≤ 0.1) [30,31]. In Equation (1), *i* is the type of pesticide, MEC adopts the measured median concentration of pesticide, and PNEC is the predicted no effect concentration. PNEC values were obtained via the ratio of the sensitive end point as the critical concentration (CC) to the acute toxicity assessment factor value (AF) though Equation (2).RQ*_i_* = MEC*_i_*/PNEC*_i_*(1)PNEC*_i_* = CC/AF(2)

In the present study, three terrestrial organisms (i.e., earthworms, springtails and mites) were used as the object of ecological risk assessment of pesticides in soil. If three no-observed effect concentration (NOEC) values are available, the AF value is 10; if two NOEC values are available, the AF value is 50; if only one NOEC value is available, the AF value is 100; if no NOEC is available, the AF value is 1000 and lethal concentration 50 (LC_50_) or effective concentration 50 (EC_50_) can be used [32].The toxicological data of a certain pesticide to terrestrial organisms were obtained from the Pesticide Properties DataBase and related references [33,34,35,36,37,38,39]. As detailed in Table 4, toxicological data of ten NEOs regarding earthworms, Collembola or springtails *F. candida* and mites are 14 days NOEC for *Eisenia fetida*, 28 days NOEC for *Folsomia candida* and 14 days NOEC for *Hypoaspis aculeifer*, respectively.

## 4. Conclusions

Systematic monitoring across China’s wheat, corn and rice agricultural soil (*n* = 69 sites) revealed ubiquitous contamination by NEOs in staple crop soils, demonstrating 100% DF overall. Compound-specific prevalence followed: imidacloprid (IMI, DF = 100%) > thiamethoxam (TMX, 88.4%) > clothianidin (CLO, 87.0%) > acetamiprid (ACE, 46.4%) > dinotefuran (DIN, 7.2%) > nitenpyram (NIT, 1.4%). The total detected NEOs concentrations exhibited crop-dependent stratification: wheat (1.77–214.55 ng/g) > corn (0.79–97.53 ng/g) > rice (0.75–72.97 ng/g). Three NEOs, IMI, CLO and TMX, dominated environmental loads, contributing over 90% of ΣNEOs across all matrices. The calculated median RQs indicated that the CLO (0.21) and TMX (0.16) in the rice soils, IMI (0.50), CLO (0.76) and TMX (0.26) in the corn soils and IMI (0.47), CLO (0.87) and TMX (0.40) in the wheat soils had a medium ecological risk (0.1 < RQs < 1). The ecological risk of the detected NEOs cannot be ignored in the soil. It is suggested to strengthen monitoring, reserve relevant pesticide remediation technologies and prevent the migration and transformation of pesticides to reduce ecological risks.

## Figures and Tables

**Figure 1 molecules-30-01803-f001:**
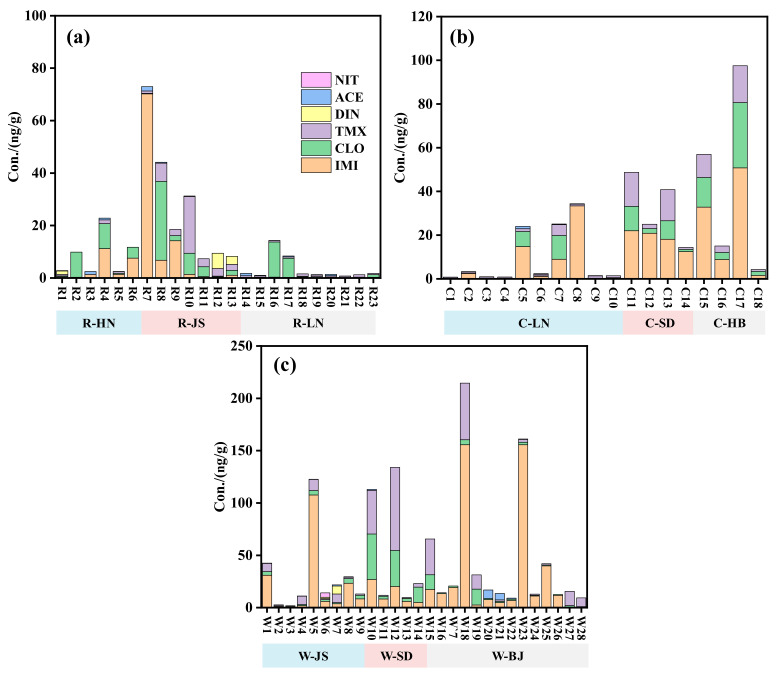
Accumulative concentrations of the detected NEOs in the (**a**) rice, (**b**) corn and (**c**) wheat field soils.

**Figure 2 molecules-30-01803-f002:**
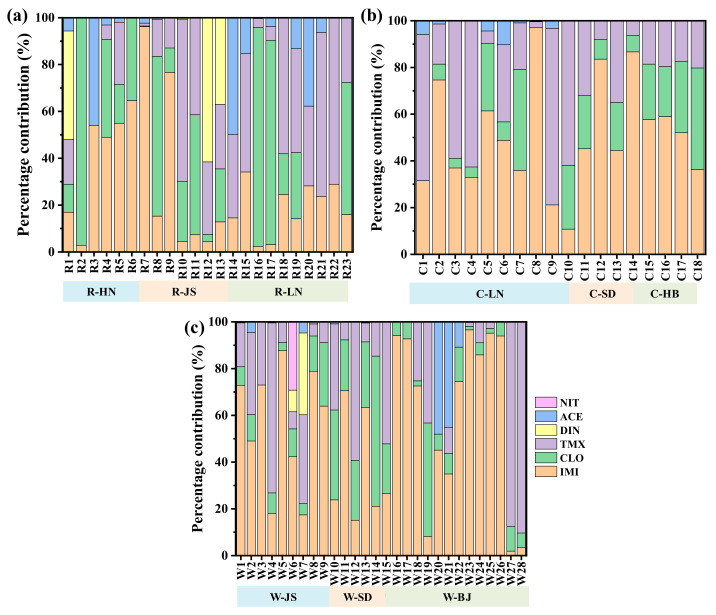
Composition distribution of the detected NEOs in the (**a**) rice, (**b**) corn and (**c**) wheat soils.

**Figure 3 molecules-30-01803-f003:**
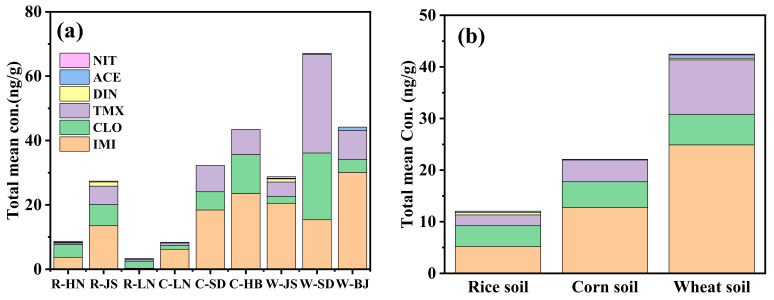
Total mean concentration of the detected NEOs in the (**a**) different provinces and (**b**) crop types of soils.

**Figure 4 molecules-30-01803-f004:**
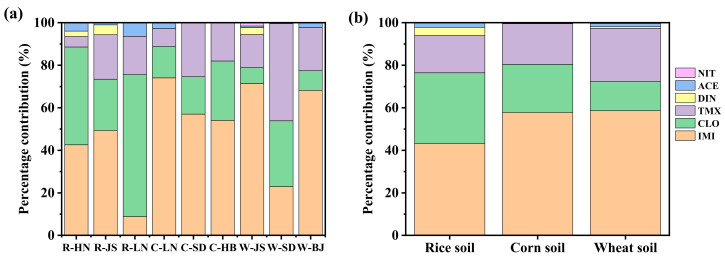
Composition distribution of the detected NEOs in the (**a**) different provinces and (**b**) crop types of soils.

**Figure 5 molecules-30-01803-f005:**
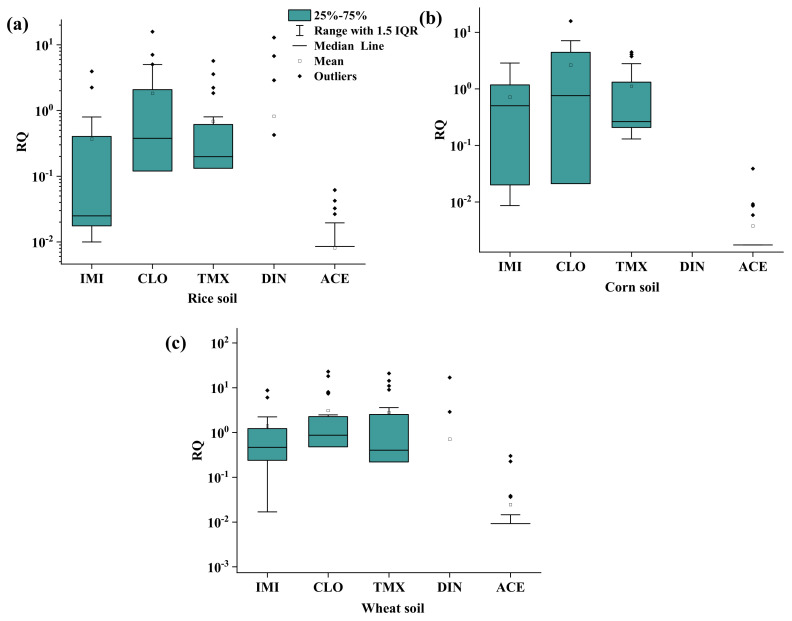
The RQs of the detected NEOs in the (**a**) rice, (**b**) corn, and (**c**) wheat soil.

**Table 1 molecules-30-01803-t001:** Detection frequency (DF) and concentration (ng/g) of NEOs in the soil samples.

Soil Type		IMI	CLO	TMX	DIN	ACE	NIT	IMTH	THIA	CYC	IPP	ΣNEOs
Rice field	DF (%)	100	78.3	87.0	13.0	56.5	0	0	0	0	0	100
	Maximum	70.20	30.09	21.63	5.81	1.66						72.97
	Minimum	0.18	0.19	0.48	<LOQ	<LOQ						0.75
	Median	0.41	0.41	0.63	<LOD	<LOQ						7.35
	Mean	5.21	4.02	2.12	0.44	0.26						12.05
Corn field	DF (%)	100	83.3	100	0	38.9	0	0	0	0	0	100
	Maximum	50.75	29.88	16.90		1.05						97.53
	Minimum	0.15	<LOQ	1.00		<LOD						0.79
	Median	8.94	1.44	4.23		<LOD						14.73
	Mean	12.76	4.99	1.85		0.10						22.08
Wheat field	DF (%)	100	96.4	82.1	7.1	42.9	3.6	0	0	0	0	100
	Maximum	155.70	43.46	79.39	7.58	8.10	4.11					214.55
	Minimum	0.30	0.30	0.48	<LOQ	<LOQ	4.11					1.77
	Median	8.32	1.65	1.52	<LOD	<LOD	<LOD					16.20
	Mean	109.61	5.86	10.60	0.30	0.66	0.15					42.47

Note: LOD: limits of detection; LOQ: limits of quantification.

**Table 2 molecules-30-01803-t002:** Comparisons of detected NEOs (ng/g) in soil samples around the world.

Location	Soil Samples	IMI	TMX	CLO	ACE	THIA	DIN	NIT	References
Belize	Crop soil Fallow soil	2.65 ± 2.42 0.118 ± 0.213	0.138 ± 0.309 0.035 ± 0.059	0.326 ± 0.728 0.199 ± 0.355	0.019 ± 0.042 0.004 ± 0.011				[12]
Philippines	Sweet peas soil Rice soil, banana, citrus soil	0.758 0.013 1.048	0.005 0.005 0.278	0.022 n.d. ^a^ 1.430	0.002 n.d. n.d.				[15]
Ontario, Canada	Maize soil, 2013 Maize soil, 2014		Σ_2 NEOs_, 4.0 ± 1.1 Σ_2 NEOs_, 5.6 ± 0.9						[14]
England	Arable soil	<0.09–10.7	<0.02–1.50	0.02–13.6					[16]
Shandong, China	Celery soil Cucumber soil Pepper soil Tomato soil	0.49–2.38 × 10^3^ 0.52–5.32 × 10^3^ n.d.-2.80 × 10^3^ 0.41–2.62 × 10^3^	n.d.-1.21 × 10^3^ 1.03–8.56 × 10^3^ n.d.-1.78 × 10^3^ n.d.-1.82 × 10^3^	0.24–2.36 × 10^3^ n.d.-3.39 × 10^3^ 3.48–1.07 × 10^3^ 0.36–1.29 × 10^3^	n.d.-2.03 n.d.-15.6 0.13–81.7 n.d.-115	n.d. n.d.-0.07 n.d.-0.07 n.d.-0.04	n.d.-35.1 n.d.-1.09 × 10^3^ n.d.-1.76 × 10^3^ 0.12–119	n.d.-9.81 n.d.-1.28 × 10^3^ n.d.-54.3 n.d.-12.8	[21]
Southern China	Citrus orchards soil	Σ_5NEOs_, 0–25.76							[22]
Tianjin, China	Land soil (spring) Land soil (fall)	0.74–1.06 × 10^3^ n.d.-2.61 × 10^3^	n.d.-1.56 × 10^3^ n.d.-2.32 × 10^3^	n.d.-74.6 n.d.-1.32 × 10^3^	0.19–4.40 × 10^3^ 0.19–31.9	n.d.-18.2 n.d.-0.14	n.d.-3.28 n.d.-1.35		[23]
Beijing, China	Wheat field soil	n.d.-5.33 × 10^3^			<1.0–1.22 × 10^3^				[5]
Six provinces, China	Rice field soil Cron field soil Wheat field soil	0.18–70.20 0.15–50.75 0.30–155.70	0.48–21.63 1.00–16.90 0.48–79.39	0.19–30.09 <0.08–29.88 0.30–43.46	n.d.-1.66 n.d.-1.05 n.d.-8.10	n.d. n.d. n.d.	n.d.-5.81 n.d.-7.58 n.d.	n.d. n.d. n.d.-4.11	The present study

^a^, n.d., not detected.

**Table 3 molecules-30-01803-t003:** The mass spectral information of ten NEOs.

Pesticides	Abbr.	Formula	R.T. (min)	Ionization Mode	Precursor Ion (*m*/*z*)	C.V. (V)	Quantitative Ion (*m*/*z*)	C.E. (V)	Qualitative Ion (*m*/*z*)	C.E. (V)
Imidacloprid	IMI	C_9_H_10_ClN_5_O_2_	2.45	ESI+	255.95	26	175.03	28	209.28	16
Clothianidin	CLO	C_6_H_8_ClN_5_O_2_S	2.36	ESI+	249.97	18	168.92	16	131.93	22
Thiamethoxam	TMX	C_8_H_10_ClN_5_O_3_S	2.19	ESI+	291.98	18	211.01	18	180.98	34
Dinotefuran	DIN	C_7_H_14_N_4_O_3_	1.10	ESI+	203.07	20	129.06	16	87.06	22
Acetamiprid	ACE	C_10_H_11_ClN_4_	2.53	ESI+	223.03	16	125.94	28	90.05	44
Thiacloprid	THIA	C_10_H_9_ClN_4_S	2.74	ESI+	252.99	20	125.99	32	98.96	54
Imidaclothiz	IMTH	C_7_H_8_ClN_5_O_2_S	2.52	ESI+	261.98	26	180.94	14	122.17	28
Nitenpyram	NIT	C_11_H_15_ClN_4_O_2_	1.08	ESI+	270.99	26	224.99	12	98.90	14
Cycloxaprid	CYC	C_14_H_15_ClN_4_O_3_	2.13	ESI+	323.05	30	276.99	14	151.01	24
Paichongding	IPP	C_17_H_23_CIN_4_O_3_	2.68	ESI+	367.11	32	137.04	26	321.05	12

**Table 4 molecules-30-01803-t004:** Toxicological data of the NEOs.

Compound	*Eisenia fetida* (mg kg^−1^)	*Folsomia candida* (mg kg^−1^)	*Hypoaspis aculeifer* (mg kg^−1^)	CC (mg kg^−1^)	AF	PNEC (mg kg^−1^)	Reference
IMI	≥0.178	1.25	>2.67	0.178	10	0.0178	PPDB
CLO	2.5	0.19 (EC_50_)	n.d.a	0.19	100	0.0019	PPDB, [33]
TMX	5.34	0.38 (EC_50_)	n.d.a	0.38	100	0.0038	PPDB, [33]
DIN	0.2	0.045 (EC_50_)	n.d.a	0.045	100	0.00045	PPDB, [34]
ACE	1.26	0.27	454	0.27	10	0.027	PPDB, [35]
THIA	0.185	10	1600	0.185	10	0.0185	PPDB, [35]
IMTH	1.41 (LC_50_)	n.d.a	n.d.a	1.41 (LC_50_)	1000	0.00141	[36]
NIT	1.32 (EC_50_)	n.d.a	n.d.a	1.32 (EC_50_)	1000	0.00132	[37]
CYC	10.21 (LC_50_)	n.d.a	n.d.a	10.21 (LC_50_)	1000	0.01021	[38]
PCD	541.07 (LC_50_)	n.d.a	n.d.a	541.07 (LC_50_)	1000	0.54107	[39]

Note: CC, critical concentration; AF, assessment factor; n.d.a, no data available; PPDB, Pesticide Properties DataBase (USDA ARS).

## Data Availability

The original contributions presented in this study are included in the article/Appendix A. Further inquiries can be directed to the corresponding author(s).

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
