# Peer review of "Occurrence and Ecological Risks of Neonicotinoids in Wheat, Corn and Rice Field Soils in China"

_molecules, 2025, doi:10.3390/molecules30081803_

Round 1
Reviewer 1 Report
Comments and Suggestions for Authors
The article contains valuable information on the fate of NEOs in crop soils. The article is very interesting and worthy of publication. Some of my comments:
Were data collected on the duration of application of these compounds - theoretically they should degrade quickly. Were other soil parameters analysed, such as temperature, moisture, granulometric composition? Perhaps the article should include a discussion of the influence of soil conditions on NEO levels. Has the content of NEOs in plants from the collected soils been investigated? The above information would be very valuable and would increase the scientific value of the publication.
Author Response
Comments: The article contains valuable information on the fate of NEOs in crop soils. The article is very interesting and worthy of publication. Some of my comments: Were data collected on the duration of application of these compounds - theoretically they should degrade quickly. Were other soil parameters analysed, such as temperature, moisture, granulometric composition? Perhaps the article should include a discussion of the influence of soil conditions on NEO levels. Has the content of NEOs in plants from the collected soils been investigated? The above information would be very valuable and would increase the scientific value of the publication.
Response: Thanks for your careful review and useful feedback.
(1) We agree with the reviewer that pesticide residue might be degraded in the soil under normal temperature in the laboratory. The collected soil samples were stored a freezer at -20℃. And then the soil samples were analyzed as soon as possible.
(2) We also agree with the reviewer that the NEOs residue in the soil had a relationship with soil parameters such as temperature, moisture and granulometric composition. Adequate soil moisture facilitates microbial proliferation and pesticide dissolution, promoting breakdown processes. However, waterlogged conditions may induce anaerobic environments that alter degradation pathways. While higher organic matter content typically increases pesticide adsorption and reduces bioavailability, it may conversely stimulate microbial activity under optimal conditions, potentially expediting decomposition. Climatically favorable regions in southern China (e.g., HN and JS) with warm or humid conditions would theoretically exhibit enhanced pesticide degradation. Nevertheless, this study reveals significantly lower NEOs residues in northern soils (e.g., LN) compared to southern counterparts. For example, the mean total concentrations of the detected NEOs in the soils from different provinces followed: JS (27.4 ng/g) > HN (8.7 ng/g) > LN (3.3 ng/g) in the rice soil, and HB (43.5 ng/g) > SD (32.2 ng/g) >LN (8.4 ng/g) in the corn soil. The observed higher residues in the south likely result from intensive pesticide use, frequent pest outbreaks, and/or soil conditions that slow degradation despite favorable climate factors. The application frequency of pesticides has a close relationship with the occurrence pattern of insect pests during the plant growing, resulting in different pesticide pollution situations in the field soils. LN province is not conducive to the benefits for the reproduction of pests compared with warm or humid conditions in southern China, resulted in less application of the NEOs. It might be one reason for the low concentration levels of the detected NEOs in the soils from LN province. The discussion has been supplemented in the revised manuscript. This change can be found in line 202-219.
(3) We agree with the reviewer that the NEOs residue in the plant is also worthy of attention. In the present study, we only focused on the NEOs residue in the soil.
Reviewer 2 Report
Comments and Suggestions for Authors
The manuscript presents an important theme that assesses the ecological risks of a group of pesticides used in various agricultural crops.
The text is well written, however some notes are necessary to improve the understanding and reading of the manuscript.
1 – Abstract
1.1 – The abstract should present the general objective of the research.
2 - Introduction
2.1 – At the end of the introduction, 3 objectives for the study are presented. The question is, what is the general objective of the research? The general objective is macro and the others should be presented in the composition of the same. Once defined, this general objective should appear in the abstract.
3 – Methods
3.1 – As observed in table 2, only the toxicity for all pesticides analyzed was presented for the organism E. fetida. Therefore, it is impossible to present an ecological risk assessment. It is worth noting that the presence of at least 3 species is essential since the susceptibility among organisms is totally variable.
3.2 - Therefore, the authors should consider removing the ecological risk assessment stage from the study.
4 – Results and Discussion
4.1 – In the topic of ecological risk assessment, the lack of data for discussion is quite evident. There is no discussion in the topic and the only point where a citation is presented for comparison is related to the aquatic environment, which is not the subject of this study (L. 256-257).
4.2 – In lines 254-256, it is suggested that there is a high ecological risk for the pesticides detected, but this information refers to only one organism, therefore it cannot be affirmed.
5 – Conclusions
5.1 – There is no doubt that the ecological risk exists, but the study did not obtain data to generate this affirmation.
6 - Based on the points raised in methods and results/discussion, it is recommended to modify the aspect of soil ecological risk assessment presented here, to relate only to the species Eisenia fetida, which is effectively the one that provides data for the study.
Comments on the Quality of English LanguageEnglish language can be improved.
Author Response
The manuscript presents an important theme that assesses the ecological risks of a group of pesticides used in various agricultural crops. The text is well written, however some notes are necessary to improve the understanding and reading of the manuscript.
Comments 1: Abstract 1.1 – The abstract should present the general objective of the research.
Response: Thanks for your comments. “The occurrence and ecological risks of ten NEOs in the field soils of three major crops (i.e., rice, wheat, and corn) in China were investigated in the present study.” as the general objective of the research has been supplemented in the abstract.
Comments 2: Introduction, 2.1 – At the end of the introduction, 3 objectives for the study are presented. The question is, what is the general objective of the research? The general objective is macro and the others should be presented in the composition of the same. Once defined, this general objective should appear in the abstract.
Response: Thanks for your comments. The general objective sentence “This study will provide a theoretical basis for paying more attention to the pollution and risk of NEOs in soil ecosystem.” has been supplemented in the Introduction.
Comments 3: Methods, 3.1 – As observed in table 2, only the toxicity for all pesticides analyzed was presented for the organism E. fetida. Therefore, it is impossible to present an ecological risk assessment. It is worth noting that the presence of at least 3 species is essential since the susceptibility among organisms is totally variable. 3.2 - Therefore, the authors should consider removing the ecological risk assessment stage from the study.
Response: Thanks for your comments. We agree with the reviewer that at least 3 species are essential since susceptibility among organisms is totally variable. In the present study, three terrestrial organisms (i.e., earthworms, springtails, mites) were used as the object of ecological risk assessment of pesticides in soil. Firstly, we have reviewed and supplemented the no-observed effect concentration (NOEC) values to ensure integrity of toxicological data for the detected NEOs (e.g., IMI, CLO, TMX, DIN, and ACE) in Table 2 [25-32]. As referred, if three NOEC values are available, the AF value is 10; if two NOEC values are available, the AF value is 50; if only one NOEC value is available, the AF value is 100; if no NOEC is available, the AF value is 1000 and lethal concentration 50 (LC50) or effective concentration 50 (EC50) can be used. When the NOEC values is not available, AF will have a response multiple to conservatively assess the soil ecological risk. Secondly, we have re-calculated and revised the risk assessment results in the manuscript in Figure 5 and section 3.2 Ecological Risk Assessment.
References
- Vasickova, J., Hvezdova, M., Kosubova, P., Hofman, J., 2019. Ecological risk assessment of pesticide residues in arable soils of the Czech Republic. Chemosphere. 216:479-487.
- Martin, W.J., Sibley, P.K., Prosser, R.S., 2023. Comparison of Established and Novel Insecticides on survival and reproduction of Folsomia candida. Environmental Toxicology and Chemistry. 42:1516-1528.
- Zhang, J., Zhang, H., Yu, C., Lin, R., Hou, Y., Li, M., Liang, H., Chen, L., Gao, X., Chen, S., 2025. Ecotoxicological effects of the neonicotinoid insecticide dinotefuran on springtails (Folsomia candida) at soil residual concentration. Pesticide Biochemistry and Physiology. 209:106345.
- Akeju, T.O.,2014. Assessment of the Effects of the neonicotinoids thiacloprid and acetamiprid on soil Fauna. Universidade de Coimbra (Portugal), Dissertation/Thesis.
- Zhang, Y., Zhang, L., Feng, L., Mao, L., Jiang, H., 2017. Oxidative stress of imidaclothiz on earthworm Eisenia fetida. Comparative Biochemistry and physiology C-Toxicology & Pharmacology. 191:1-6.
- Ge, J., Xiao, Y., Chai, Y., Yan, H., Wu, R., Xin, X., Wang, D., Yu, X., 2018. Sub-lethal effects of six neonicotinoids on avoidance behavior and reproduction of earthworms (Eisenia fetida). Ecotoxicology and Environmental Safety. 162:423-429.
- Zhang, J., Xiong, K., Chen, A., Li, F., 2016. Toxicity of a novel neonicotinoid insecticide paichongding to earthworm eisenia fetida. Soil and Sediment Contamination.26:235-246.
- Qi, S., Wang, D., Zhu, L., Teng, M., Wang, C., Xue, X., Wu, L., 2018. Effects of a novel neonicotinoid insecticide cycloxaprid on earthworm, Eisenia fetida. Environmental Science and Pollution Research.25:14138-14147.
Comments 4: Results and Discussion, 4.1 – In the topic of ecological risk assessment, the lack of data for discussion is quite evident. There is no discussion in the topic and the only point where a citation is presented for comparison is related to the aquatic environment, which is not the subject of this study (L. 256-257). 4.2 – In lines 254-256, it is suggested that there is a high ecological risk for the pesticides detected, but this information refers to only one organism, therefore it cannot be affirmed.
Response: Thanks for your comments. We have deleted the inappropriate literature, cited the related references and conducted in-depth discussions in the manuscript. Yu et al. [37] found that the current residual levels of NEOs in the soils of an agricultural zone within the Pearl River Delta in South China could pose sub-lethal or acute effects to non-target terrestrial organisms such as earthworms. The application of IMI in peanut fields treated with flowable concentrate for seed treatment formulations, it predicted low to moderate earthworm toxicity and a medium risk from ecotoxicity exposure [38]. Wu et al. [39] reported the ecological risks of ACE and TMX were higher than other NEOs in the soils of tomato and cucumber greenhouses. Consequently, the ecological risks posed by pesticide residues in agricultural soils exhibit significant variation across different cropping systems. This underscores the critical need for enhanced environmental monitoring, with particular emphasis on NEOs in soil ecosystems.
References
- Yu, Z., Li, X., Wang, S., Liu, L., Zeng, E., 2021. The human and ecological risks of neonicotinoid insecticides in soils of an agricultural zone within the Pearl River Delta, South China. Environmental Pollution. 284:117358.
- Abdul, K., Wu, C., Man, Y., Liu, X., Dong, F., Zheng, Y., 2024. Residue behavior of imidacloprid FS formulation in peanut cultivation system in china and its dietary and ecological risk assessment. Environmental Geochemistry and Health. 47:35.
- Wu, R., He, W., Li, Y., Li, Y., Qin, Y., Meng, F., Wang, L, Xu, F., 2020. Residual concentrations and ecological risks of neonicotinoid insecticides in the soils of tomato and cucumber greenhouses in Shouguang, Shandong Province, East China. Science of the Total Environment. 738:140248.
Comments 5: Conclusions, 5.1 – There is no doubt that the ecological risk exists, but the study did not obtain data to generate this affirmation.
Response: Thanks for your comments. We supplemented the data to generate this affirmation for ecological risk in Conclusions. The calculated median RQs indicated that the CLO (0.21) and TMX (0.16) in the rice soils, IMI (0.50), CLO (0.76), and TMX (0.26) in the corn soils, and IMI (0.47), CLO (0.87), and TMX (0.40) in the wheat soils had a medium ecological risk (0.1<RQs<1).
Comments 6: Based on the points raised in methods and results/discussion, it is recommended to modify the aspect of soil ecological risk assessment presented here, to relate only to the species Eisenia fetida, which is effectively the one that provides data for the study.
Response: Thanks for your comments. We have supplemented the NOEC values and then revised the risk assessment results. As referred, when the NOEC values is not available, AF will have a response multiple to conservatively assess the soil ecological risk. The ecological risk assessment has been modified and discussed in the revised manuscript section 3.2 Ecological Risk Assessment.
Round 2
Reviewer 2 Report
Comments and Suggestions for Authors All requests were accepted and adjustments were made.The manuscript is therefore suitable for publication.